# Risk Factors for Neurological Deficits Following Brain Tumor Resection in the Supplementary Motor Area (SMA): A 66-Case Double-Center Study

**DOI:** 10.3390/cancers17081369

**Published:** 2025-04-19

**Authors:** Lucio De Maria, Karl Schaller, Daniel Kiss-Bodolay, Giuseppe Barbagallo, Jibril Osman Farah

**Affiliations:** 1Neurosurgery Unit, Department of Clinical Neurosciences, Geneva University Hospitals (HUG), 1205 Geneva, Switzerland; karl.schaller@hcuge.ch (K.S.); daniel.kiss-bodolay@hcuge.ch (D.K.-B.); 2Neurosurgery Unit, Department of Medical, Surgical Sciences and Advanced Technologies “GF Ingrassia” (DGFI), University of Catania, 95124 Catania, Italy; gbarbagallo@unict.it; 3Neurosurgery Unit, The Walton Centre NHS Foundation Trust (WCFT), Liverpool L9 7LJ, UK; jibril.farah@thewaltoncentre.nhs.uk

**Keywords:** supplementary motor area, SMA syndrome, gliomas, metastases, surgery, neurological deficits, postoperative deficits, case series

## Abstract

Surgical resection of tumors involving the supplementary motor area (SMA) often leads to a transient clinical syndrome characterized by motor and/or language deficits. In this retrospective study of 66 SMA-involving tumor resections from two centers, we identified key clinical, functional-anatomical, and intraoperative factors associated with these postoperative deficits. Pre-existing medical conditions were linked to a higher risk of language impairments. The proximity of the tumor to the corticospinal tract and functional MRI (fMRI) activation patterns were associated with motor deficits, while the absence of intraoperative language mapping increased the risk of language deficits. Our findings highlight the importance of a multimodal approach—combining preoperative fMRI and tractography with intraoperative neurophysiological monitoring—to optimize the extent of resection while preserving neurological function. These strategies should be routinely employed to guide safe surgical planning in patients with SMA tumors.

## 1. Introduction

The supplementary motor area (SMA) is a cortical region situated in the posterior segment of the superior frontal gyrus [1,2]. It occupies the medial aspect of Brodmann area 6 (AB6), with the premotor area (PMA) occupying the lateral aspect. The superior frontal sulcus serves as the demarcation line between the two regions [3,4]. The SMA is both anatomically and functionally divided into two distinct regions, namely the pre-SMA and the SMA-proper. The pre-SMA is located in the anterior region, while the SMA-proper is situated in the posterior region [5]. As shown in Figure 1, these two regions are conventionally separated by an imaginary vertical line, known as the vertical commissure anterior (VCA) line, which passes through the anterior commissure and runs perpendicular to another imaginary line connecting the anterior and posterior commissures [6]. While the SMA-proper is mainly involved in motor functions such as generating and controlling movements, the pre-SMA is primarily implicated in cognitive functions [7,8,9].

Surgical resection or damage of the SMA can result in a negative motor response. This response was first defined as “SMA syndrome” by Laplane et al. [10]. This syndrome has mainly been reported as a consequence of brain tumor resections or resections for epilepsy [11,12,13]. However, similar symptoms have also been reported for infarcts of the medial frontal lobe [14]. The severity of the syndrome can be complete or partial. In complete SMA syndrome, there is complete contralateral hemiplegia, with or without mutism. In partial SMA syndrome, there is contralateral hemiparesis and/or speech hesitancy [15]. An important characteristic of SMA syndrome is the preservation (or slight reduction) of muscle tone compared to the hypertonic response observed in corticospinal tract (CST)-related injuries. The assessment of reflex excitability and muscle tone in the early postoperative evaluation and later at follow-up can help differentiate SMA hypokinesia from other motor deficits.

The syndrome progresses in three stages. The first stage is characterized by global akinesia and language disturbances, depending on the involvement of the dominant hemisphere. The second stage shows sudden improvement of symptoms a few days later. The final stage is characterized by almost undetectable sequels weeks or months later [16]. The neurological impairment associated with SMA syndrome is generally transient, with patients experiencing a remission of symptoms in a few weeks. This is mainly due to plasticity and compensatory mechanisms that involve both contralateral and residual ipsilateral SMAs [17,18,19]. Notably, transitory neurological deficits, including those associated with SMA syndrome, are a well-recognized phenomenon in onco-functional neurosurgery. These temporary impairments, which may involve various cognitive and motor domains, are often considered functional complications rather than permanent outcomes. Such deficits are thought to arise from temporary disruptions in neural circuits due to functional surgeries aiming to maximize resection while minimizing lasting impairments. The early recovery observed in these cases highlights the importance of distinguishing these deficits from permanent injuries, as the latter are associated with significantly worse long-term outcomes.

It has been reported that the risk of neurological deficits after resection in the SMA can vary widely from 23% to 100% [20,21]. Several factors can influence this risk, including age, comorbidity, disease recurrence, preoperative neurological status, type of tumor, and extent of resection [22,23,24]. Some authors have reported that preoperative functional magnetic resonance imaging (fMRI) or transcranial magnetic stimulation (TMS), as well as intraoperative direct cortical stimulation (DCS), may predict the occurrence of neurological deficits and recovery [25,26]. However, tumors in the SMA region are relatively infrequent. Thus, the evidence for surgical treatment of such tumors is primarily based on small retrospective case studies conducted in a single medical center. Additionally, previous studies have focused on only a few variables, which has limited our knowledge of patients’ outcomes after tumor resection. To gain a better understanding of the risk of neurological deficits after tumor removal in the SMA region, we conducted a study of clinical, preoperative, and intraoperative variables in a double-center cohort.

## 2. Materials and Methods

We reviewed data from more than 100 patients diagnosed with brain tumors in two medical centers since 2007. The patients included in our study had cerebral lesions involving the SMA and were diagnosed with either glioma or metastasis based on a histopathological examination. We excluded World Health Organization (WHO) grade 1 gliomas from the study.

Baseline demographics, clinical presentation, tumor side, histopathology, preoperative imaging, intraoperative functional monitoring, and clinical and imaging follow-up data were collected.

### 2.1. Preoperative Imaging

Patients underwent contrast-enhanced computed tomography (CT) and magnetic resonance imaging (MRI) scans before surgery. The preoperative MRI protocol included the following sequences: T1-weighted (T1W), T2-weighted (T2W), fluid-attenuated inversion recovery (FLAIR), diffusion tensor imaging (DTI), gadolinium-enhanced T1-weighted (Gd-T1W), arterial spin labeling (ASL) and dynamic susceptibility contrast (DSC) perfusion-weighted imaging (PWI), and spectroscopy (MRS). Additionally, functional MRI (fMRI) was performed in 26 cases, and DTI-derived CST tractography was carried out in 43 cases.

#### 2.1.1. Functional Magnetic Resonance Imaging (fMRI)

Functional images were acquired through single-shot echo planar imaging (SS-EPI) with a repetition time (TR) of 3000 ms, a time to echo (TE) of 35 ms, a flip angle of 90°, and a field of view (FOV) of 20 × 20 cm. The BrainWave Real-Time software (ver. 2.0; GE Healthcare, Waukesha, WV, USA) was used to control the transmission of stimuli, acquire responses, and display images. The patients were asked to perform motor and/or verbal paradigms. The motor paradigms consisted of 6 cycles, each involving a 15 s rest phase followed by a 15 s action phase. During the action phase, patients performed pre-set tongue tasks, hand tasks, and foot tasks. The verbal paradigms comprised 6 cycles, each including a 20 s rest phase and a 40 s action phase. Patients performed pre-set rhyming, word generation, or verb generation tasks. The raw functional data were processed using the BrainWave Post-Acquisition software (ver. 2.0; GE Healthcare, Waukesha, WV, USA) and included a data quality check, motion correction, temporal filtering, and spatial smoothing. Finally, the functional activation maps were fused with high-resolution pre-task T1W or T2W sequences.

The fMRI images were uploaded to the neuronavigation station, allowing intraoperative recognition of the eloquent areas. For this study, the minimum distance between the area of primary motor cortex (PMC) functional activation and the tumor was measured as shown in Figure 2. Each activation located on the central sulcus was considered to represent the PMC. Common anatomical landmarks such as the omega sign (“hand knob”) were used to identify the central sulcus on axial images. Among different motor tasks, only those that evoked a cluster of activation nearest to the tumor were considered significant. Similarly, if more than one cluster of activation was evoked by the same motor task, only the one closest to the tumor was considered.

#### 2.1.2. Diffusion Tensor Imaging (DTI)-Derived Tractography

Diffusion images were acquired through SS-EPI with a TR of 6599–8280 ms, a TE of 76 ms, and an FOV of 224 × 224 mm. Images were processed using the software StealthViz with the module StealthDTI (ver. 1.2, Medtronic, Minneapolis, MN, USA). Color-coded fractional anisotropy (FA) maps were created from DTI. The CST was reconstructed on the side of the tumor using multiple cubic regions of interest (ROIs), as shown in Figure 3. Tractography was performed using the iPlan Cranial software (ver. 3.0.5; Brainlab, Feldkirchen, Germany) with an FA threshold of 0.2 and a minimum fiber length of 40 mm.

The resulting CST images were uploaded to the neuronavigation station and helped with the intraoperative recognition of the sub-cortical motor fibers. For this study, the minimum distance between the CST and the lesion was measured on the sagittal view.

### 2.2. Surgery

The surgical imaging plan was uploaded to the neuronavigation station for intraoperative guidance. A total of 61 surgeries were conducted under general anesthesia, while 5 were conducted under conscious sedation through loco-regional anesthesia (awake surgeries). Awake surgeries were performed at various time points throughout the study period, with no apparent correlation between timing and surgical indications. The decision to proceed with an awake surgery was not solely based on anatomical and functional considerations but was also guided by a comprehensive preoperative neuropsychological evaluation. This assessment ensured that awake mapping would be both feasible and beneficial, taking into account the patient’s cognitive and linguistic profile, as well as their psychological readiness to tolerate the procedure. Ultimately, the decision required the patient’s full agreement through informed consent.

#### Intraoperative Neurophysiology Monitoring

Intraoperative functional monitoring was utilized to assist in the surgical procedure of 45 cases. In 22 of those cases, a 4-contact platinum recording strip electrode was used to register somatosensory-evoked potentials (SSEPs) directly from the cortex during peripheral nerve stimulation. The phase reversal of the SSEPs helped locate the PMC and central sulcus.

In all 45 cases, motor-evoked potentials (MEPs) were registered, and the resection was functionally limited. In 43 of those cases, the cortex was directly stimulated (DCS) with a monopolar probe as an anode and a sub-dermal needle electrode placed in the scalp at the frontal midline electrode position (Fz) as a cathode. In 15 cases, subcortical stimulation was also performed by reversing the polarity of the monopolar probe (cathode) and sub-dermal needle electrode (anode).

Pairs of sub-dermal needle EMG electrodes were placed in the opposite musculature to record MEPs (Appendix A). Stimulation was performed using a short train of rectangular pulses at a high frequency, following the method described by Taniguchi et al. [27]. The stimulation frequency was set at 250 Hz, with a pulse duration of 0.5 ms and a train number of 6 pulses. This technique produced muscle responses at a much lower intensity of charge compared to the traditional method of eliciting movement of the extremities by applying a train of pulses at a lower frequency (50–60 Hz). The stimulus intensity varied depending on the region stimulated as follows: 5–20 mA for the PMC, 12–25 mA for the SMA, and 5–25 mA for subcortical stimulation.

In 9 cases, the same sub-dermal needle EMG electrodes were used to record MEPs while continuously stimulating the cortex with a 4-contact strip electrode as an anode and a sub-dermal needle electrode placed in the scalp at the Fz as a cathode. The stimulation was performed at a rate of 1.0 Hz. Once a stable baseline of stimulation was established, a 50% amplitude reduction criterion was used to identify potential motor impairment. In 16 cases, the sub-dermal electrodes were also used to record spontaneous EMG, and no spontaneous discharges were observed in any of the patients.

In 5 cases, awake surgery allowed intraoperative language mapping, and the resection was limited to preserve function. The cortex was directly stimulated while the patient was performing verbal tasks guided by a neuropsychologist. Language disturbances such as slurring, stuttering, or speech arrest during stimulation were considered indicators of the language cortex.

EEG activity was registered during all cortex stimulations from a 4-contact platinum strip electrode, looking for discharges, spikes, and slow wave anomalies. EEG activity was also used to assess the depth of sedation.

### 2.3. Postoperative Evaluation

For this study, we have categorized the postoperative neurological status based on the classical presentation of SMA syndrome. Motor symptoms have been classified into three grades of severity: (A) normal status, where motor skills were entirely preserved, (B) hemiparesis with variable degrees of unilateral weakness, and (C) hemiplegia. Similarly, verbal symptoms have been classified into three grades of severity: (A) normal status, where verbal skills were intact, (B) dysphasia with speech disturbances of variable degrees, and (C) aphasia, where speech capacity was completely abated. Based on the combination of motor and language symptoms and their reversibility, the neurological status has been classified into partial SMA syndrome (hemiparesis and/or speech hesitancy), complete SMA syndrome (hemiplegia with or without mutism), and permanent neurological deficit (any motor or verbal symptoms without remission over the following weeks/months).

The postoperative MRI protocol included T1W, T2W, FLAIR, diffusion-weighted imaging (DWI), and Gd-T1W. We measured the residual tumor volume on axial Gd-T1W sequences using the iPlan Cranial software (ver. 3.0.5; Brainlab, Feldkirchen, Germany).

### 2.4. Statistical Analysis

The study investigated the relationship between immediate postoperative neurological deficits and preoperative, intraoperative, and clinical risk factors using Fisher’s exact probability test. A statistical significance of *p* < 0.05 was considered. In addition to the overall analysis, we conducted separate subgroup analyses for primary (gliomas) and secondary (metastatic) tumors to account for their distinct characteristics and potential differences in risk factors.

## 3. Results

### 3.1. Patient Population

A total of 53 patients, 33 males and 20 females, were included. The mean age was 44. Patients were divided into four groups depending on the histopathological diagnosis: the first group includes WHO grade 2 gliomas (14 patients), the second group WHO grade 3 gliomas (10 patients), the third WHO grade 4 glioblastomas (21 patients), and the last one includes metastasis (8 patients). Eleven patients were re-operated because of progression/recurrence after the first surgery. Concerning those re-operated patients, only the last diagnosis was considered for histopathological classification, as shown in Appendix A. If all surgeries are considered separately, the total number of cases included in the study is 66. The tumor location was the right side in 32 cases and the left side in 34 cases. In order of frequency, the symptoms of presentation were the following: seizures (35 cases), motor deficits (21 cases), language deficits (9 cases), cognitive impairments (4 cases), sensory deficits (3 cases), and headache (11 cases). Appendix A and Figure 4 summarize baseline population data.

Appendix A summarizes fMRI and tractography data, while Appendix A summarizes intraoperative neurophysiology data of the patient population. A clear phase reversal was detected in 19 cases. Responses in SMA were recorded in only seven cases, and the minimum threshold was 20 mA. No clear difference in stimulation thresholds was found between patients who were asleep and patients who were awake, though a larger awake sample would be needed for better comparison. In two cases, an abnormal EEG activity was detected: in case n° 4, the patient experienced a seizure following language mapping, and a propofol bolus was given; in case n° 11, an intermittent seizure-like activity was recorded during the waking period that abated once the patient was awake.

### 3.2. Outcomes

The postoperative outcomes are presented in Figure 5 and Appendix A. In 23 cases, a motor deficit was observed during the immediate postoperative period. The severity of the motor deficit was grade B (hemiparesis) in 22 cases and grade C (hemiplegia) in one case. However, all these patients experienced a remission of their symptoms within the following weeks or months. In 13 cases, a language deficit occurred, with the severity being grade B (dysphasia) in 12 cases and grade C (aphasia) in one case. Except for one patient, all of them experienced a remission of their symptoms within the following weeks or months. Postoperative rehabilitation, including physiotherapy and/or speech therapy, was provided according to each patient’s specific recovery needs.

Overall, 42% of patients (28 cases) developed a postoperative neurological deficit. In 26 cases, the symptomatology was classified as partial SMA syndrome, while one patient developed complete SMA syndrome, and one patient developed a permanent neurological deficit, specifically a language rather than a motor deficit.

### 3.3. Risk Analysis

In our analysis, we studied how clinical, preoperative, and intraoperative risk factors might affect the development of postoperative motor and language deficits.

Appendix A present the analysis of clinical risk factors. We studied age, clinical history, recurrence of disease, tumor grade, and preoperative symptomatology. We found a statistically significant correlation (*p* = 0.005) between familial or personal comorbidities and the development of postoperative language deficits. This correlation remained significant only for gliomas in the subgroup analysis (*p* = 0.001).

Appendix A present the analysis of preoperative risk factors. We analyzed the fMRI activation of the SMA in response to motor tasks, the dominant hemispheric activation in response to language tasks, the minimum distance between the area of PMC fMRI activation and the tumor, and the distance between the CST and the tumor. We found a statistically significant correlation (*p* = 0.044) between the fMRI activation of the SMA in response to motor tasks and the development of motor deficits. We also found that the distance between the CST and the tumor was significantly correlated (*p* = 0.005) with the development of motor deficits. Only the latter correlation remained significant for gliomas in the subgroup analysis (*p* = 0.019).

Finally, Appendix A present the analysis of intraoperative risk factors. We examined the monitoring of MEPs, the language mapping, and the extent of resection. We found a statistically significant correlation (*p* = 0.044) between the lack of intraoperative language mapping and the development of postoperative language deficits. We also found that the extent of resection was significantly correlated (*p* = 0.040) with the development of postoperative language deficits. However, only the correlation between lack of intraoperative language mapping and language deficits remained significant for gliomas in the subgroup analysis (*p* = 0.049).

## 4. Discussion

Several clinical, preoperative, and intraoperative variables were analyzed to identify the risk factors for the development of transient or permanent postoperative neurological deficits in patients undergoing a tumor resection in the SMA. Our overall analysis revealed that the patient’s medical history, intraoperative language mapping, and extent of resection significantly influenced the occurrence of language deficits. Furthermore, the proximity between the CST and the tumor, and the fMRI activation of the SMA in response to motor tasks were found to correlate with the development of motor deficits. However, we did not find any correlation between the lack of intraoperative monitoring of MEPs and the development of motor deficits.

Given the distinct pathophysiological characteristics of primary (gliomas) and secondary (metastatic) tumors, we performed a subgroup analysis to better understand potential differences in risk factors. While gliomas are known for their infiltrative nature and the gradual adaptation of functional networks through neuroplasticity (particularly lower-grade ones), metastases primarily displace rather than infiltrate the surrounding brain tissue. This difference may have significant implications for both surgical strategy and functional recovery, as compensatory mechanisms could vary between tumor types. Additionally, in metastatic cases with edema or significant mass effect, the accuracy of preoperative fMRI and tractography may be affected, potentially influencing clinical correlations.

Despite the sample-related limitations, this stratified approach revealed that correlations between familial or personal comorbidities and language deficits, as well as between the lack of intraoperative language mapping and language deficits, remained significant only in the glioma subgroup. Similarly, in this subgroup, tumor proximity to the CST correlated with motor deficits. However, no significant associations were found between the extent of resection and language deficits or between fMRI activation of the SMA in response to motor tasks and motor deficits. Likewise, in the metastasis subgroup, no statistically significant correlations were observed, likely due to the smaller sample size, in addition to the pathophysiological differences discussed above.

These findings highlight the need for larger patient cohorts in future studies to better understand the differences between these two tumor subgroups and to refine surgical strategies aimed at minimizing postoperative neurological deficits.

### 4.1. Clinical Factors

This study found that among the clinical variables, only the past medical history had a relevant impact on the outcome. Although the patient’s age is known to increase the operatory risk, no link was found between age and the occurrence of postoperative motor or language deficits. A significant correlation was found between past medical history and the development of postoperative language deficits, but not motor deficits. This may be explained by the fact that language function relies on a broader and more complex network of brain regions compared to primary motor output. While higher-order cognitive functions likely require greater neural integration, potentially increasing susceptibility to interference from pre-existing conditions, no measurable effect of medical history was observed on basic motor performance. However, our study did not investigate which specific conditions may pose a higher risk. Further research would be valuable in exploring the influence of cognitive reserve and relevant medical conditions on higher cognitive domain outcomes.

Contrary to what was reported by Ibe Y et al. [23], no association was found between the recurrence of disease and postoperative neurological deficits. Although subsequent operations should be associated with a major resection of eloquent areas, many other factors might be involved in preventing the occurrence of neurological deficits even when the risk of their development is higher. Foremost among these additional factors is intraoperative neurophysiology monitoring, which allows a functionally limited resection, permitting the real-time recognition of eloquent areas during surgery. Additionally, tumor grade was not found to be a significant risk factor for postoperative deficits in our analysis. This is particularly intriguing, as high-grade tumor patients are often considered more vulnerable to the effects of SMA syndrome due to the potential impact on their overall quality of life. Given the need for adjuvant therapy, these patients may struggle with recovery if a transient deficit occurs, as fatigue and limited rehabilitation opportunities could hinder functional improvement.

Finally, the study analyzed preoperative symptomatology. While presenting motor or language symptoms is an indicator of tumor localization in the vicinity of eloquent areas and a higher risk of postoperative functional impairment, many other factors are involved. For instance, peritumoral edema and intratumoral necrosis are both reasons why the resection may improve the neurological status of the patient. After comparing the preoperative neurological status with the immediate postoperative outcome of the case studied population, the study found that in 38 cases, there was an improvement in the symptomatology, but in as many as 28 cases, the clinical picture worsened. However, SMA syndrome is known to be a benign and reversible syndrome, and 98.5% of the population had a remission of the symptomatology within the following weeks or months. Therefore, if we compare the preoperative symptomatology with the later postoperative symptomatology, we find that only in one case was there a real worsening of the clinical picture because the patient developed permanent language deficits. A permanent motor deficit is attributable to the damage of the PMC or the fibers of the CST, while a permanent language deficit is ascribable to damage to other cortical or subcortical regions essential for speech.

### 4.2. Preoperative Factors

Among the preoperative variables, we found that the fMRI activation of the SMA in response to motor tasks and the distance between the CST and the tumor significantly correlated with the development of postoperative motor deficits. This suggests that fMRI is a valuable tool to predict a patient’s motor outcome. However, the study also found inter-individual variability in the population, where sometimes only the PMC is activated in response to motor tasks instead of both the SMA and PMC. This variability could bring about different responses to a resection of the SMA. Regarding language tasks, we expected that the ipsilaterality between the tumor and the dominant hemispheric fMRI activation would correlate with the development of postoperative language deficits. However, no statistically significant correlation was found for this variable. It is essential to be cautious about this conclusion as the study did not investigate the particular anatomical relationship between the tumor and language areas. Additionally, the sample size for this variable was very small (eight patients, with five having ipsilateral activation and three with contralateral activation). While speech disturbances are more commonly associated with left-sided SMA surgeries due to the typical left-hemispheric dominance for language, reports suggest that right-sided SMA lesions can also lead to transient language impairments, particularly affecting speech initiation and fluency [6,7]. This may be attributed to the bilateral involvement of the SMA in speech production and its role in higher-order motor control.

This study also investigated the correlation between the distance of the tumor from the PMC and CST and the immediate motor outcome. We expected that neither of these variables would correlate with the immediate motor outcome, as damage to the PMC or CST fibers typically results in permanent rather than transient motor deficits. However, a statistically significant correlation was found for the CST–tumor distance variable, suggesting that even minor manipulation of CST fibers can contribute to transient motor dysfunction. This finding indicates that while SMA syndrome is primarily attributed to cortical disruption, subcortical fiber involvement may also play a role in certain cases. It further reinforces the notion that CST fibers are particularly susceptible to intraoperative manipulation, and their proximity to the tumor may influence the risk of transient postoperative deficits.

### 4.3. Intraoperative Factors

Among the intraoperative variables, language mapping and extent of resection were found to be significantly related to the neurological outcome of the patients, particularly the language outcome. Regarding the intraoperative MEP monitoring, we assumed that patients who underwent surgeries without the aid of neurophysiology monitoring are at higher risk of developing postoperative SMA syndrome. A *p*-value of 0.100 resulted, arguing against the previously mentioned hypothesis (Appendix A). This verdict can be understood if we consider that no response in the SMA was obtained under a stimulus intensity of 20 mA, as reported in Appendix A. An MEP evoked by direct cortical stimulation of the SMA at an intensity greater than 20 mA is more likely to be due to the spread of the stimulus to the PMC and CST fibers rather than to the actual response of the SMA. This means that this procedure was not able to detect the real eloquence of the SMA, consequently not preventing its resection. Conversely, a correlation was found between intraoperative language mapping and the development of postoperative language deficits. However, this analysis considers just five cases with intraoperative language mapping, compared to 61 cases with no mapping. The two samples are too different, and therefore this result should be regarded cautiously. While the absence of intraoperative language mapping was associated with a higher risk of language deficits, it is also important to consider that traditional intraoperative mapping approaches may not fully capture the complex functional roles of the SMA and pre-SMA. These regions contribute not only to motor and language functions but also to cognitive and behavioral processes, including motor planning, bimanual coordination, and multitasking. Moreover, postoperative SMA syndrome has been reported even in cases where intraoperative mapping failed to identify positive sites, suggesting that deficits may result from network disconnection rather than direct cortical resection [16,19]. This highlights the potential need for refined mapping techniques, such as awake paradigms with lower-frequency stimulation, to better identify functionally relevant regions. Further studies are needed to optimize intraoperative strategies for preserving SMA function and to better understand the interplay between resection, functional compensation, and neuroplasticity.

Finally, the extent of resection was evaluated as a potential risk factor for both language and motor deficits. We assumed that patients who underwent gross total resection (GTR) would be more likely to develop postoperative neurological deficits. While no statistically significant correlation was found with motor deficits, a significant association with language deficits emerged in the overall analysis, with 13 patients experiencing postoperative language impairment—10 following subtotal resection (STR) and 3 following GTR. However, this correlation did not remain significant in the subgroup analysis. It should be noted that the extent of resection is often the result of a surgical decision aimed at balancing oncological radicality and functional preservation. Therefore, the observed association may partly reflect the surgeon’s decision to interrupt resection based on intraoperative neurophysiological findings, which may have prevented the occurrence of more severe deficits. This highlights how the mere categorization of resection might be misleading, as the anatomical features of the resection cavity and the involvement of surrounding cortical and subcortical structures are critical for interpreting these results. Notably, Ibe et al. [23] found that a larger resection of the posterior part and/or the medial wall of the SMA-proper was associated with a higher rate of postoperative motor deficits. Further research incorporating residual tumor volume alongside the anatomical characteristics of the resection cavity would be valuable, given its prognostic significance and potential impact on postoperative outcomes.

### 4.4. Surgical Decision-Making in SMA Tumor Resections

A critical aspect of SMA tumor surgery is the balance between maximizing oncological resection and minimizing functional impairment. In our institutions, the decision-making process was guided by preoperative imaging and functional assessments, intraoperative neurophysiological monitoring, and the expected reversibility of SMA syndrome. While postoperative SMA syndrome is typically transient, the extent of resection should be carefully planned to avoid irreversible deficits, particularly when the tumor extends toward the CST or critical language areas.

Preoperatively, the surgical strategy is tailored based on the tumor’s anatomical location, functional imaging results, and the patient’s overall condition. If the tumor is touching critical structures such as the CST, a more cautious approach is adopted to prevent permanent deficits. In contrast, in cases where the tumor is confined to the SMA without significant proximity to subcortical motor pathways, a more extensive resection is often pursued, considering the high likelihood of functional recovery from SMA syndrome.

Intraoperatively, neurosurgeons rely on direct cortical and subcortical stimulation to define functional boundaries. When stimulation at low intensities elicits motor responses or disrupts language tasks, resection is halted to preserve function. Additionally, real-time monitoring of MEPs influences surgical decisions, ensuring that resection does not extend into non-redundant functional areas.

Ultimately, our surgical philosophy is not to systematically avoid SMA syndrome but rather to achieve the best possible balance between oncological radicality and functional preservation. While SMA syndrome is generally reversible, we recognize that temporary deficits can significantly impact patients’ quality of life, particularly in those undergoing adjuvant therapy. Therefore, surgical decisions are highly individualized, with a nuanced approach to maximizing tumor removal while minimizing long-term functional morbidity.

### 4.5. Limitations

The major limitation of this study is the lack of a larger sample of patients harboring tumors in the SMA who developed a permanent neurological deficit postoperatively. For this reason, it was not possible to evaluate the relationship between the diverse clinical, preoperative and intraoperative variables and the later postoperative outcome. Indeed, Fisher’s exact test was applied only concerning the immediate postoperative outcome, i.e., when SMA syndrome is still manifested.

Moreover, the subgroup analysis of primary (gliomas) and secondary (metastatic) tumors was affected by sample size limitations, particularly in the metastasis subgroup. While this stratified approach allowed us to explore differences between these tumor types, the smaller samples may have limited statistical power and prevented the identification of significant correlations.

Furthermore, the size and precise anatomical location of the tumor were not systematically analyzed as potential risk factors. A more detailed volumetric and anatomical assessment of the tumors in relation to the SMA-proper and pre-SMA could offer further insights into the likelihood of postoperative deficits.

Additionally, the use of deterministic tractography (i.e., iPlan Cranial software) represents an intrinsic limitation of the study. Deterministic tractography may have reduced accuracy in accounting for inter-tract distances, particularly in cases involving edema or infiltrative FLAIR-positive tumors, which can alter tractography results and impact the analysis of functional connectivity.

Nonetheless, to our knowledge, this is the most comprehensive risk analysis to date, evaluating the broadest range of variables associated with SMA syndrome and immediate neurological deficits following tumor resection in the SMA.

## 5. Conclusions

The surgical resection of the SMA is associated with the development of a well-characterized, reversible clinical syndrome known as SMA syndrome, which includes motor and/or language deficits of varying severity. Patients with pre-existing medical conditions may be at a higher risk of impairments in higher cognitive domains, such as those involved in language function. Tractography and fMRI may help predict the risk of motor deficits, while intraoperative language mapping can aid in preventing postoperative language deficits. The extent of resection, along with the anatomical characteristics of the resection cavity, correlates with postoperative outcomes. Traditional intraoperative mapping approaches may not fully capture the complex functional roles of the SMA and pre-SMA. Further intraoperative studies are warranted to determine the most effective paradigms for mapping and monitoring tumors involving these regions.

To ensure the best outcomes for patients with SMA tumors, a combined approach using preoperative and intraoperative neuroimaging techniques such as fMRI and DTI, along with neurophysiological monitoring procedures like intraoperative cortical and subcortical stimulation, should be routinely implemented.

## Figures and Tables

**Figure 1 cancers-17-01369-f001:**
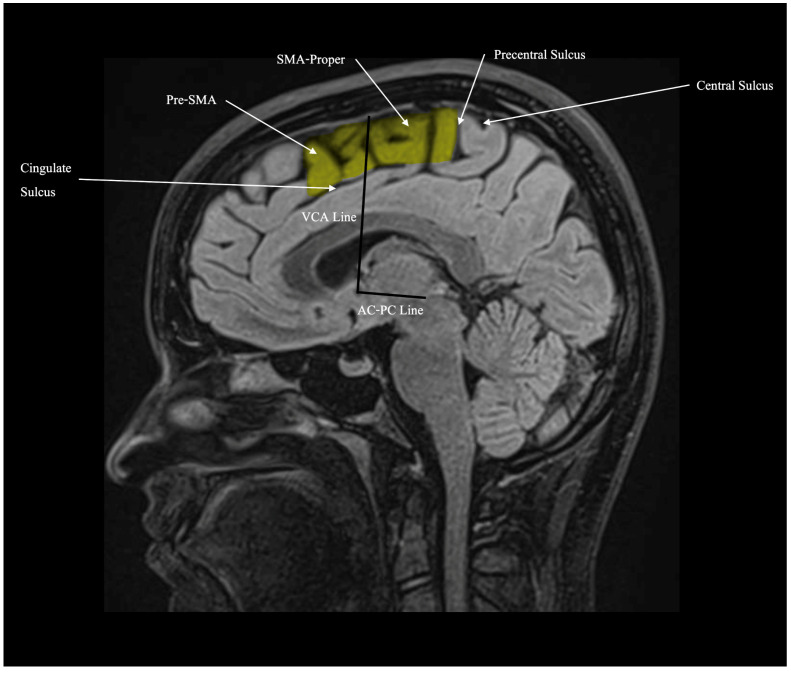
The SMA is highlighted in yellow. It is divided into the pre-SMA (anterior) and the SMA-proper (posterior) by the VCA line, which passes through the anterior commissure and runs perpendicular to the anterior commissure–posterior commissure (AC–PC) line.

**Figure 2 cancers-17-01369-f002:**
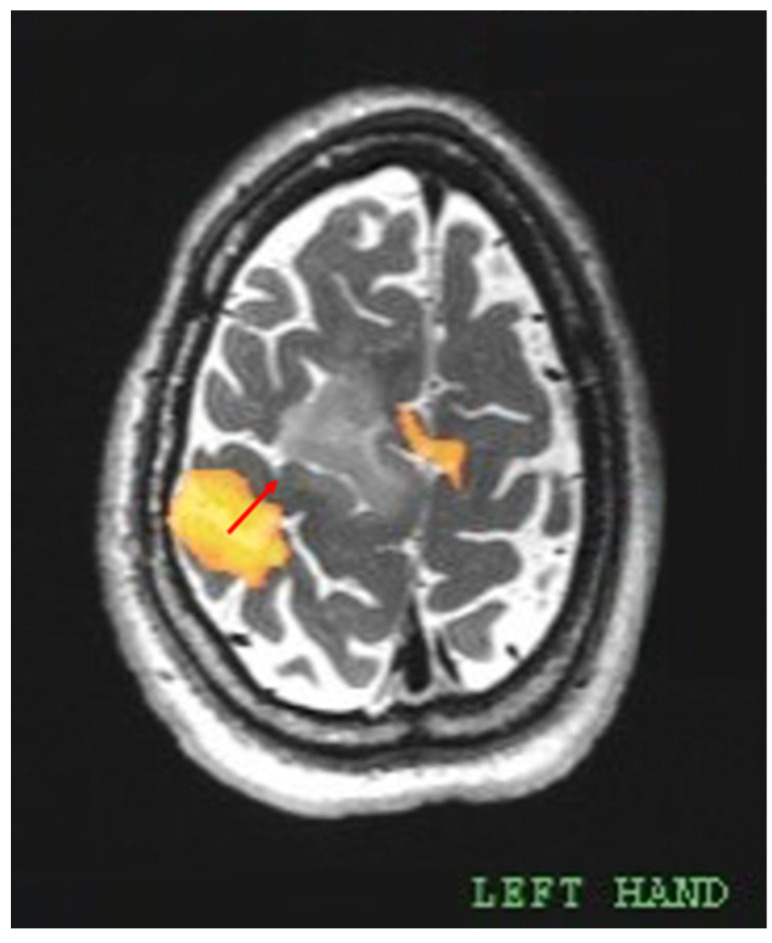
The distance was measured between the center of mass (COM) of the activation cluster, which was evoked by a left-hand motor task at the PMC, and the edge of tumoral signal abnormality. The resulting distance was found to be 18.72 mm (red arrow). Another cluster of activation, located anteromedial to the tumor, was identified as the SMA. Carestream Vue PACS software (ver. 11.4.1.0324, Carestream Health, Rochester, NY, USA).

**Figure 3 cancers-17-01369-f003:**
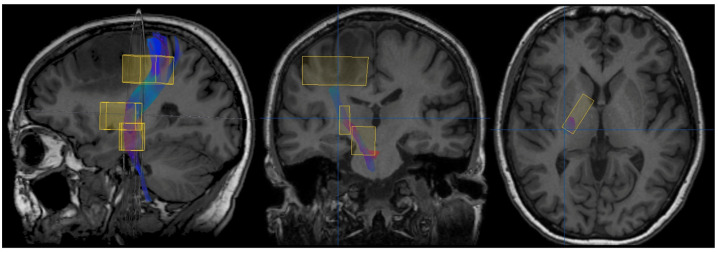
CST tractography. The first ROI was placed at the pre-central gyrus, the second ROI at the posterior limb of the internal capsule, and the third ROI at the cerebral peduncle. iPlan Cranial software (ver. 3.0.5, Brainlab, Feldkirchen, Germany).

**Figure 4 cancers-17-01369-f004:**
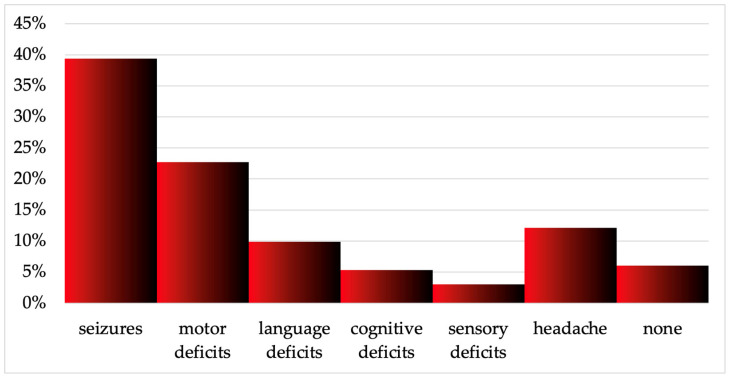
Presenting symptoms of the patient population. Six percent of patients manifested no presenting symptoms, and their diagnosis was incidental.

**Figure 5 cancers-17-01369-f005:**
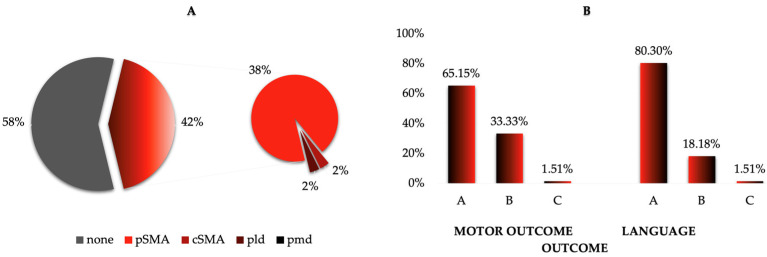
Postoperative presentation. (**A**) Of all the patients, 38% developed pSMA, 2% developed cSMA, and 2% a permanent neurological deficit. However, 58% of patients had no symptoms after the operation. (**B**) As for the motor outcome, 33% of patients had hemiparesis (grade B), while only 2% had hemiplegia (grade C). Most patients (65%) did not have any motor issues. Concerning the language outcome, 18% of patients had dysphasia (grade B) and 2% had aphasia (grade C). The majority of patients (80%) had no language disturbances.

## Data Availability

Data are contained within the article and Appendix A.

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
