# Peer review of "Risk Factors for Neurological Deficits Following Brain Tumor Resection in the Supplementary Motor Area (SMA): A 66-Case Double-Center Study"

_cancers, 2025, doi:10.3390/cancers17081369_

Round 1

Reviewer 1 Report (Previous Reviewer 1)

Comments and Suggestions for Authors

I sincerely thank the authors for the methodological revisions provided here and the extended discussion reformulated accordingly. 

At the current stage, I have no further corrections in mind that should halt the publication of the manuscript in this reviewed form. 

This manuscript is a resubmission of an earlier submission. The following is a list of the peer review reports and author responses from that submission.

Round 1

Reviewer 1 Report

Comments and Suggestions for Authors

The study proposes a detailed examination of the risks associated with neurological deficits following surgical resection of brain tumors in the Supplementary Motor Area (SMA) in a retrospective cohort of 53 patients across two centers. The study focuses on several factors, including clinical history, preoperative imaging, and intraoperative neurophysiology, seeking to correlate these with postoperative outcomes, specifically the so-called SMA syndrome. The study incorporates advanced imaging techniques such as fMRI and DTI, along with intraoperative monitoring (i.e. MEPs monitoring) and language mapping, aligning with modern studies in the onco-functional field. 

Here, I provide an organised opinion about the manuscript’s sections:

Introduction:

The introduction is well structured. However, I suggest mentioning muscle tone together with other features of SMA syndrome. Muscle tone and reflex excitability can be tested in the early post-operative evaluation and later at follow-up as a specific characteristic for distinguishing a corticospinal tract-related motor deficit from SMA hypokinesia. Indeed, muscle tone preservation (or slight reduction) compared to hypertonic response in pyramidal injury, might be worth a mention. 
Also, I would provide additional space to detailing transitory deficits in onco-functional procedures, where a early recovery it is a well known phenomenon in many major cognitive domains other than being specific for SMA and pre-SMA areas. Despite not being fully understood, several authors argue about considering transitory deficits as functional complications rather than temporary consequences for functionally-limited surgeries to achieve the highest rate of resection with no permanent deficits as the latter, and not temporary disturbances, are associated with a worse outcome in most previous studies. 

Methods:

Given the use of a deterministic tractography (I.e., iPlan Cranial software), I would disclose the intrinsic limitation of such a method when accounting for inter-tract distances overall and in cases where edema or infiltrative FLAIR positive tumour are present. 

Lines 161-162 present a repetition. Please correct. 

Results: 

Including secondary tumours in number 8 cases might require an additional section in the discussion and – perhaps – further analysis. Even though I wouldn’t like to encourage a stratification given the limited sample, we might acknowledge that metastases tend to behave as tumours dislocating the surrounding brain parenchyma and fibers instead of infiltrating over time. The latter aspect bears both surgical and functional considerations to be accounted for: first, in conditions where edema and mass effect are present, we can mainly argue about the accuracy of fMRI and tractography, other than clinical presentation. Moreover, as they are not infiltrative diseases, the brain-tumour interaction might differ from gliomas (lower grade primarily, but also high grade as reported in previous literature), and function compensation and neuroplasticity might act in distinguished fashion, posing an internal bias when evaluating these tumours all together. For this reason, I suggest adding separate analyses (for primary and secondary tumors), being aware of the sample-related limitations in both directions. 

Among 61 surgeries, five were conducted with an awake protocol, despite 34 left-sided tumours. Although no simplistic statement can be made on the indication for an awake surgery—as all onco-functional surgeries should be tailored to patients’ characteristics and needs—it would be helpful to detail the reasons for such a skewed indication in favour of asleep surgeries. Also, as the study interval was notably wide, a review of the trend over time might be informative. 

The risk analysis and surgery-related outcomes might be further discussed. For example, the authors correctly acknowledge the association between permanent language deficits and the absence of awake brain mapping for such a procedure. Similarly, the lack of positive SMA sites during motor mapping should not be straightforwardly interpreted as negative sites overall. As described in the literature, intra- and post-operative SMA syndrome development is reported after disconnection/resection of previously negative DES sites, where high-frequency monopolar stimulation was applied. Additional discussion can be provided here on the need for different paradigms: as the authors mentioned in the introduction, SMA and pre-SMA bear cognitive, complex motor (among all, multi-task coordination, bimanual performance, etc) behaviour, and others. Accordingly, an awake paradigm with low frequency stimulation might show behavioural disruption in previously negative sites. I would acknowledge this in the limitations, as further studies are required to weight and optimise the use of IOM and brain mapping in the premotor cortex, especially in the medial prefrontal cortex. Nevertheless, neuroplasticity and recovery of temporary deficits are to be evaluated, and the best balance between radicality and functionality addressed, regardless of the technique used. 

Again, I would acknowledge in lines 235-241 whether or not there was a difference in stimulation threshold between asleep and awake patients. 

Discussion:

Line 299-300: I would instead not label motor control as “low-order activity.” Indeed, what was directly tested or monitored was direct pyramidal activation in terms of motor output, not the whole motor performance. As complex motor behaviour was not investigated, I would omit such a statement or reformulate referring to additional cognitive domains with high complexity - not included in the investigation -which might require several different regions involved as the language network. It would be interesting to speculate on the weight of cognitive reserve and previous medical histories on higher cognitive domain outcomes after brain disconnections. As it is beyond the aim of the study, I would clarify that previous relevant history seems to affect only complex domains, and no effect was measured in simple primary motor performance.

Line 374-382: I would focus more on reviewing the results. According to the statistics reported, 14 patients experienced a language deficit postoperatively, of which 10 had a STR and 3 a GTR (with a significance at p=0.04). As the authors disclosed, the mere categorization of resection might be misleading, and the actual surgical cavity and surrounding cortical areas and white matter tracts should be considered when interpreting these results. However, it would be of interest for the reader to further review the characteristics of these patients experiencing post-operative language deficits. Moreover, the inclusion of residual tumour volume is warranted, given its prognostic value. Conclusions: 

According to the previous comments, I might also suggest remodeling some concepts in the conclusions. Further intraoperative studies are warranted to understand the best paradigm to deploy for fruitful mapping and monitoring of tumours harbouring SMA and pre-SMA. 

Final comment:  The study – other than the comments above – is well conducted and focuses on a high-yield topic for the current scientific community; moreover, the limitations are well addressed at the end of the manuscript. Despite these limitations and some revisions I suggested, the study makes a fair contribution to the field by detailing the complex interplay of factors influencing the risk of neurological deficits post-SMA resection. It emphasizes the critical role of intraoperative language mapping in preventing language deficits and highlights how fMRI and tractography might predict the risk of motor deficits. These findings advocate for a personalized approach to SMA surgery, which is widely accepted.  In conclusion, although the study provides valuable insights into the risk factors and outcomes associated with SMA surgery in neurooncology, its impact is somewhat tempered by methodological constraints and revisions are required. 

Reviewer 2 Report

Comments and Suggestions for Authors

Dear authors,

this article focuses on the risk factors of developing motor and language deficits after SMA tumors surgery. It identifies some preoperative and intraoperative predictive factors that can help the surgeon fully inform the patients. This study is of high interest because the series gathers imaging and clinical information. 

Comments: 

- Did you expect patients with right-sided SMA surgery to develop speech disturbance as often as left-sided ? Could you discuss this point with references ? 

- After reading this article, it is still not clear which past medical history elements are risk factors for developing SMA syndrom. l.267 "We found a statistically significant correlation (p=0.005) between familial or personal comorbidities and the development of postoperative language deficits." Does any preexisting medical condition, in the family or the patient, increase the risk ? This result sounds a bit vague, since questioning of the patient, and of his family history, is prone to much uncertainty. What did you consider past medical history ? 

- l272 " the minimum distance between the area of PMC fMRI activation and the tumor, and the distance between the CST and the tumor". Is it possible to establish a threshold associated with a high risk of developing a deficit? How can we use this result in practice ? 

- "the extent of resection was significantly correlated with the development of deficits" could you detail this result ? Why were some surgery interrupted before complete resection was performed ? Was it to avoid a deficit, that could be predicted by intraoperative testing for instance? If that's the case, this result is not relevant, since it is self-predicted by the surgeon's decision to interrupt surgery

- Do you have any data about the risk as evaluated by the surgeon before surgery ? 

- Is the size of the tumor or its precise location correlated with the SMA syndrome ? The authors make a great analysis of fMRI and tractography, but we lack mere anatomical details about the tumors. In addition, a figure locating SMA would be helpful for readers not fully familiar with the anatomical region.

- l353 "suggesting that manipulation of CST fibers can cause even a mere transient dysfunction..." Do you suggest that SMA syndrome is due to CST fibers manipulation rather than cortical SMA lesion in some cases ? Could you explicit this point ?

- How do you explain one patient develop a complete definitive motor deficit ? Could it be predicted a posteriori ? Could it be avoided ? Did any vascular event happen that could explain the deficit ? 

- The authors find no correlation between the grade of the tumor and the prognosis. This is an intriguing result + we usually consider that patients with high grade tumors should be avoided SMA syndrome because their quality of life will be highly impaired in case of developing a transient deficit + they are usually too tired to perform physiotherapy at the same time as their adjuvant treatment. Could the authors discuss this point ? 

- After reading the whole study, it is globally unsatisfying that the reader cannot fully understand 1) how predictable the SMA syndrome was, based on the previous literature, and 2) how the neurosurgeon decided to stop the surgery to avoid SMA or even CST associated-syndrome. Could you detail in the discussion the role of the surgeon in making this decision preoperatively when planning the surgery, and intraoperatively. What was the surgeon's state of mind in your hospitals? Do you try to prevent SMA syndrome or do you try to achieve complete resection in all cases, since you know SMA syndrome will be reversible ? Depending on this philosophy, the extent of resection will be different. 

Minor comments: 

- l.251", only a minority of patients (28 cases) developed a postoperative neurological deficit" doesn't sound appropriate since this is still a high proportion of patients (add%).

- Table 1 could be a supplemental table.

- Figure 4 needs improving, in terms of presentation (grey background, small text...)

- I wished the results were presented in a figure rather than the text only

- All patients probably had postoperative rehabilitation (physiotherapy or speech therapist), could you precise that, and evaluate how many months they needed it before being back to their normal neurological examination?